# Mammal Roadkills in Lithuanian Urban Areas: A 15-Year Study

**DOI:** 10.3390/ani13203272

**Published:** 2023-10-19

**Authors:** Linas Balčiauskas, Andrius Kučas, Laima Balčiauskienė

**Affiliations:** 1Laboratory of Mammalian Ecology, Nature Research Centre, Akademijos 2, 08412 Vilnius, Lithuania; laima.balciauskiene@gamtc.lt; 2Territorial Development Unit (B3), European Commission, Joint Research Centre, Via Fermi 2749, 21027 Ispra, Italy; andrius.kucas@ec.europa.eu

**Keywords:** wild mammals, domestic mammals, roadkill, urban areas, COVID-19, anthropause

## Abstract

**Simple Summary:**

We examined traffic accidents involving wildlife and domestic animals in Lithuanian urban areas from 2007 to 2022. We analyzed the proportions of wild and domestic animals in roadkill, annual trends, the dominant species involved, and monthly changes during the COVID-19 periods of mobility restrictions. During the study period, the number of roadkills increased exponentially, with roe deer (*Capreolus capreolus*) being the most numerous species. The proportion of domestic animals, 12.2% of the total roadkill in urban areas, significantly exceeded that on non-urban main, national, and regional roads in the country, these being 3.8%, 3.6%, and 4.3%, respectively. During the 2020 and 2021 COVID-19 restrictions, there was a significant increase in the number of accidents involving animals in cities, which again normalized after the lockdowns. Thus, mobility restrictions had only a temporary effect. The increase in animal-related accidents, even when there were fewer people on the roads, suggests that new ways of protecting animals in cities might be required, at least through improving driver awareness on the issue.

**Abstract:**

We investigated roadkills in urban areas in Lithuania from 2007 to 2022, including two periods with COVID-19 restrictions on people’s movement. We analyzed the proportions of wild and domestic animals in roadkill, annual trends, the predominant species involved, and monthly changes during the restrictions. Urban roads were characterized by a low species diversity of roadkilled mammals, with roe deer (*Capreolus capreolus*) dominating. Total numbers increased exponentially during the study period. The proportion of domestic animals, 12.2%, significantly exceeded that on non-urban roads in the country. The proportion of domestic animals decreased from over 40% in 2007–2009 to 3.7–5.4% in 2020–2022, while the proportion of wild mammals increased from 36.1–39.6% to 89.9–90.6%, respectively. During the periods of COVID-19 restrictions, the number of roadkills in urban areas was significantly higher than expected based on long-term trends. Compared to 2019, the number of roadkilled roe deer in 2020–2021 almost doubled from 700 to 1281–1325 individuals. These anthropause effects were, however, temporary. The imbalance between the roadkill number and transport intensity might require new mitigation strategies to sustain mammal populations in urban areas, at least through improving driver awareness on the issue.

## 1. Introduction

Urban growth, or urbanization, is a long-term and one of the most important drivers of biodiversity loss [1,2]. Therefore, understanding urban biodiversity is important for its conservation [3], as urbanization continues to increase [4] and urban pressures are expressed differently on ecological communities or species [5,6]. Among the questions requiring attention are the suitability of urban landscapes for different species, spatiotemporal dynamics of these species, and the response of these species to technological changes [7]. The last question became important during the COVID-19 pandemic, when imposed restrictions of human and transport mobility created an effect termed “anthropause” [8].

Anthropause has a certain degree of similarity with the post-industrial transformation of cities in developing wild areas and is characterized by restoration of wilderness [9]. Abandoned land in declining urban areas, if converted into green spaces, benefits ecosystem health, wildlife species and, of course, people [10]. Cities may support many plant and animal species [11], with some of them capable of increasing in numbers as compared to non-urban habitats.

Mammals are not equally represented in different cities, a phenomenon known as the “luxury effect”, which states that bigger cities are less suitable for mammal species and that urban intensity is the strongest factor [12]. The presence of green areas such as urban parks, residential gardens, and remaining scrubland is important for mammals [13]. Connectivity with surrounding habitats is also very important for the abundance of mammal species in urban areas [14], and mammals require certain behavioral changes to survive in urban environments [15].

In Lithuania, knowledge of the mammal fauna is limited to the species composition of the country’s largest city, Vilnius [16]. A total of 51 species have been recorded in Vilnius, with 29 species recorded in the most urbanized part of the city [17]. However, the data cover a period of 20 years ago, and in the future can be supplemented with new observation methods such as wildlife cameras [18].

The presence of wildlife along transport routes almost inevitably leads to collisions between animals and vehicles, despite all the measures taken to mitigate them [19]. Urban areas have higher road densities; therefore, even small populations are at high risk of road mortality [20]. In suburbs and other urban–wildlife interface areas, the migration of animals, especially ungulates, increases the risk of these collisions [21]. The intensity of wildlife–vehicle collisions (WVC) is influenced by differences in land cover, traffic volumes, and mitigation measures; however, the relationship between WVC and traffic intensity is not the same for different mammalian species, being unimodal (negative linear, positive linear) or U-shaped [22].

Road accident studies show that the number of WVCs in urban and rural areas can be similar, despite significant differences in traffic volume [23]. When comparing an urban landscape with a protected area, a decrease in traffic intensity towards the latter was associated with higher wildlife mortality due to roadkills [24]. The number of road mortalities in a peri-urban area has been found to be species dependent in relation to traffic intensity [25], and was habitat related.

Domestic animals are generally not dominant in animal–vehicle collisions, and thus are only analyzed in a few studies. For example, in a roadkill study in Brazil, domestic animals accounted for 34% of the total [26], and in Korea, a proportion of 27.0% was found [27]. In Tanzania, domestic animals accounted for only 6.25% of roadkilled mammals [28]. In Lithuania, roadkills of domestic animals were not included in the animal–vehicle collision analysis in more than 15 publications. These species have been excluded from the sample analyzing the diversity of roadkilled mammals [29], or were not separated in the sample analyzing COVID-19-related WVC issues on the country’s main, national, and regional roads [30].

The impact of the anthropause and COVID-19 regulations on biodiversity has primarily occurred in urban areas. The evidence has confirmed that the reduction in human activities has a direct and strong impact on the environment [31]. Many wildlife species have become more active in urban environments, but the increased contact with animals is thought to be due to changes in human behavior [32]. It remains unclear whether animals have actually returned to urbanized areas, or whether there has just been an increase in sightings [33]. Nevertheless, the COVID-19 pandemic was a new opportunity to study urban environments and wildlife [34].

Research in Chile has shown an increase in sightings of carnivores in urban environments [35], unexpected impacts of human activities on adaptive species such as wild boar (*Sus scrofa*) in peri-urban habitats near Prague in the Czech Republic [36], or white-tailed deer (*Cervus nippon*) in the Nara Park in central Japan [37]. Although the range of movement and road avoidance behavior was highly individual, it was assumed that landscape permeability is higher in tightly locked areas characterized by limited human movement [38]. Therefore, the number of roadkilled animals is expected to change. In Spain, the number of roadkills on local roads has increased since 2020, when it was minimal due to movement restrictions, and domestic animals were affected [39].

In Lithuania, a significant decrease in roadkill numbers on main, national, and regional roads, especially after the start of the first period of COVID-19 mobility restrictions and the start of the second period of restrictions, April–May 2020 and November–December 2020, respectively, was reported [30]. The aim of this study was to analyze mammal roadkills in urban areas, which include the streets of cities, towns, smaller settlements, and the roads that pass through these areas, in 2007–2022, namely, including the period of COVID-19 restrictions on mobility. Specifically, we analyze the proportions of wild and domestic animals in roadkill, the dominant species, annual trends, and monthly changes during periods of restrictions. This is the first study of its kind in Lithuania and the Baltic States.

## 2. Materials and Methods

### 2.1. Study Site

An investigation on roadkills was carried out in Lithuania, for the period 2007–2022, covering main, national, and regional road network and urbanized territories (Figure 1). Lithuania has 21,238 km of roads, out of these, 1750.71 km are main roads with an annual average daily traffic (AADT) about 10,000 vehicles/day, 4927.68 km are national roads, with an AADT up to 3000 vehicles/day, and 14,559.24 km are regional roads with an AADT up to 500 vehicles/day [40,41]. We focused mostly on the roads, in which road classification and AADT data are not available. These are roads passing through urban areas (cities and suburbs), and streets. No WVC reduction measures have been implemented on these roads.

The most common land type in Lithuania is agricultural land (52.6%), with forests accounting for another 33.2%, built-up territories 3.64% and roads 1.61%. The human population density is 45.3 inhabitants per square km [42,43].

**Figure 1 animals-13-03272-f001:**
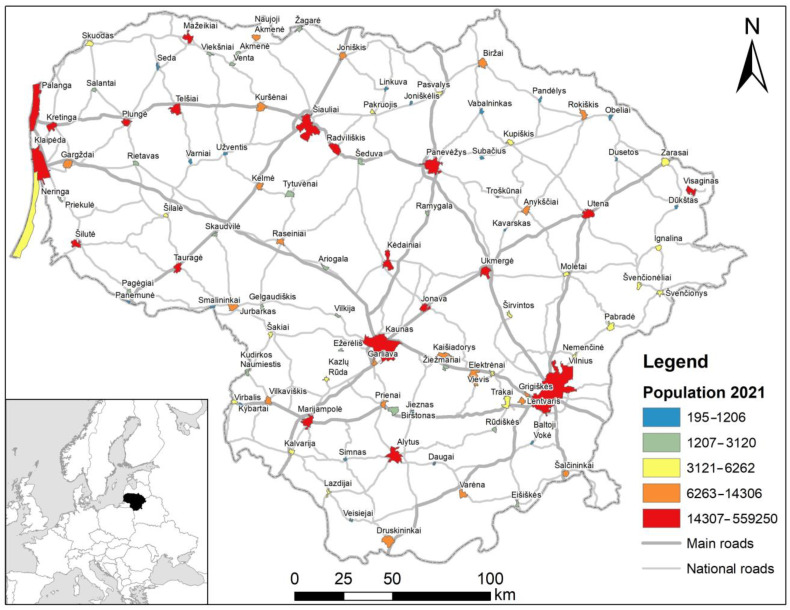
The main and national road network in Lithuania and the position of cities and towns (urbanized territories). Regional roads are not shown. Urbanized territory boundaries according to [44,45] and https://open-data-ls-osp-sdg.hub.arcgis.com/, accessed on 19 August 2023.

### 2.2. Roadkill Data

We used two data sources: data on the roadkill provided by the Lithuanian Police Traffic Surveillance Service and data collected by the authors from the Nature Research Centre. In the first case, WVCs are reported by drivers and recorded by the police, and in the second case, the data are collected by professional biologists who drive in search of roadkill. Between 2007 and 2022, 3730 registrations were made during 3670 registration sessions on a 283,000 km route. Both data sources include 47,554 roadkills of domestic and wild mammals. Appendix A contains maps of mammal roadkills (Figure A1) and roe deer roadkills (Figure A2) in Lithuania during the study period, 2007–2022.

WVC fatalities on main, national, and regional roads were georeferenced, but WVC fatalities in urbanized areas were not. In the last case, the location of a roadkill may be indicated by the street and number, the name of a village or settlement, etc. Therefore, the urban roadkill maps below do not cover all points and do not analyze habitat associations. The main source of data in urbanized areas is the Lithuanian Police Traffic Surveillance Service.

### 2.3. Lockdown Periods

According to the Re-open EU website [46], there were two COVID-19 restriction periods in Lithuania. The first restriction period lasted 112 days, from 26 February 2020 to 17 June 2020, and the second period lasted 240 days, from 4 November 2020 to 2 July 2021. People were asked to refrain from leaving their place of residence except for everyday needs (food shopping, work-related matters, and health matters). Travel between municipalities was also restricted, except for essential needs, and residents were required to own real estate at the destination. Thus, although the police controlled the movement of transport between municipalities, it was not completely banned. At the end of each restriction period, the controls were lifted and the restrictions eased.

Between the first and second restriction period, recommendations were made to restrict the movement of people, but transport movements were not controlled. After the end of the second restriction period, no further re-restrictions were imposed.

There are no numerical estimations of traffic counts in urban territories during the restricted periods. Bates et al. reported a 36% reduction in average driving time in Lithuania [47].

### 2.4. Data Analysis

Diversity of roadkilled mammals was assessed by the number of species (S), Shannon’s H index calculated with base *e*, and the dominance (D) index according to Krebs [48] and Hammer et al. [49]. Dominance was expressed as a 1-Simpson index, ranging from 0 (all taxa equally distributed) to 1 (one taxon completely dominates the community). Pairwise comparisons between H and D were made using the t-criterion, and the bootstrap method with *n* = 999 replicates was used to calculate the variance [50].

For proportions expressed as percentages, we used Fisher’s 95% CIs as confidence intervals, and the effect size was estimated using Cohen’s w-criterion [51], both calculated using WinPepi version 11.39 software [52].

Using regression analysis, we estimated the annual and monthly dynamics of roadkills. The best regression was chosen using the least squares method and the coefficient of determination R^2^. In all cases, the empirical distribution of roadkills was best described by a power regression. Based on regression, posterior predicted annual and monthly numbers were obtained.

The significance of the difference between the observed and predicted numbers was assessed using the chi-square statistic and the Wilcoxon signed rank test (W) for equivalence of means. A confidence level was set at *p* < 0.05. Calculations were performed with PAST, version 4.13 (Paleontological Museum, University of Oslo, Oslo, Norway).

## 3. Results

Between 2007 and 2022, a total of 47,554 mammals from 40 species were killed in road collisions in Lithuania. Among these species, eight were domestic mammals, while the remaining 32 were wild ones (refer to Table 1 for details).

In total, domestic mammals accounted for 5.4% (95% CI = 5.2–5.7%) of the observed cases. Their prevalence was most pronounced on urban roads, constituting 12.2% (95% CI = 11.5–12.9%), surpassing their occurrence on main roads (3.8%, 95% CI = 3.5–4.1%), national roads (3.6%, 95% CI = 3.4–3.9%), and regional roads (4.3%, 95% CI = 3.9–4.8%). These disparities were statistically significant with *p* < 0.0001, reflected in corresponding χ^2^ values of 584.0, 723.6, and 310.6.

However, it is noteworthy that the effect size of domestic mammal roadkills, when comparing urban roads with the main, national, and regional roads, was small. The calculated Cohen’s w-values were 0.160, 0.165, and 0.138, respectively.

Only a few species are more likely to be killed on urban roads than on other types of roads—main, national, or regional. All of these species—horse, goat, rabbit, and dog—were domestic animals (Table 1).

In Vilnius, the capital of Lithuania, a large proportion of mammal roadkills are associated with major roads (A1, A1, A3, A4, A14, A15, A16, and A19) exiting the city (Figure 2). In Kaunas, the second largest city, most accidents occur on the A1 and A5 main roads. Collisions with mammals have also been recorded in these towns and smaller outgoing national roads, as well as on the streets. In the smaller cities of Klaipėda and Panevėžys, there were fewer collisions with mammals, and the significance of outgoing main roads was less significant (Figure 2).

The patterns of deer collisions in the four urban areas were similar (Figure 3), with the most important in the largest cities being the outgoing main roads with the highest traffic volume. However, some of the data for these towns are not geocoded; therefore, no habitat analysis has been carried out.

### 3.1. Diversity of Roadkilled Mammals by Road Type, 2007–2022

In terms of the diversity of roadkilled mammals, urban roads exhibit similarities to those of regional significance. The similarity is indicated by comparably low species counts, low diversity indices, and high dominance proportions (Table 2). Notably, dominance levels on both urban and regional roads were contingent upon the prevalence of the most represented species, specifically the roe deer, which constituted 68.6% and 69.9% of the respective totals.

In contrast, main roads demonstrated distinctive characteristics. They registered the highest count of species and displayed greater diversity, while concurrently displaying a lesser degree of dominance. This is attributed to the fact that the most represented species, the roe deer, comprised a mere 38.7% of the overall count on these roads.

National roads exhibited a comparable number of roadkilled mammal species (for a detailed species list, refer to Table 1). However, their diversity was reduced compared to main roads, while their dominance patterns remained akin to those observed on urban roads, as indicated in Table 2.

These distinctions were not contingent on the sample size, as shown by the species accumulation curves. The differences are demonstrated at an approximate sample size of 1000 specimens for species counts (Figure 4a), and only around one hundred specimens for diversity estimations (Figure 4b). Consequently, the aforementioned observations also hold true for the examination of annual variations in roadkills on urban roads, where the sample size ranged from 238 to 3238 individuals spanning the period from 2007 to 2022. Similarly, the same principles apply to monthly variations, with a sample size of 608 to 1119 roadkill incidents per month.

The dominant mammal species across all types of roads was the roe deer. On urban roads, two other frequently encountered roadkill species were the domestic dog, accounting for 10.0%, and the moose, contributing to 4.0% of cases. Similarly, on main roads, the raccoon dog was featured as the second most frequently killed species, constituting 9.5%, followed by the moose at 7.4%. On national roads, the moose (5.4%) and the wild boar (4.3%) were prevalent species, while regional roads mirrored the same pattern, with these species comprising 3.7% and 4.6%, respectively.

### 3.2. Annual Numbers and Diversity of Roadkilled Mammals on Urban Roads, 2007–2022

The annual increase in the number of mammals roadkilled in urban areas is best approximated by an exponential regression that explains 93% of the variation (Figure 5a). The number of animals killed on roads in the COVID-19 years, 2020 and 2021, was by 55.0% and 29.4% more than expected, respectively. In numbers, 943 more roadkills than expected were observed, and this increase was significant (χ^2^ = 10.8, *p* < 0.005). In 2022, the number of roadkills decreased and became close to that predicted by the regression. 

A similar pattern was observed for roe deer roadkills (Figure 5b), as the exponential regression explained 96% of the variation in numbers by year. The excess of roadkills of roe deer observed in 2020 and 2021 was 64.8% and 23.0%, respectively, compared to the predicted number. During 2020 and 2021, 761 more roe deer were roadkilled than expected (χ^2^ = 23.02, *p* < 0.001). In 2022, the number of observed roe deer roadkills decreased and was 8.8% lower than the forecast.

In total, domestic animals roadkilled in urbanized areas accounted for 12.2% (95% CI = 11.5–12.9%), wild mammals for 78.8% (95% CI = 78.0–79.7%), and 9.0% of the roadkills were of unknown species. The share of domestic animals has been decreasing: 40% and more in 2007–2009, 17.4–37.6% in 2010–2015, and 11.3–14.4% in 2016–2019. In the COVID-19 years, the share of domestic animals in roadkill decreased to 4.3% in 2020, 5.4% in 2021 and, later, accounted for 3.7% in 2022. In contrast, the proportion of wild animals in roadkill increased to 89.9–90.6% in 2020–2022 from only 36.1–39.6% in 2007–2009. The proportion of unidentified cases decreased from a peak of 20.9% in 2009 to between 5.6 and 5.9% in 2020–2022.

Of the five most represented species, the dynamics of roadkills of roe deer and domestic dogs are controversial (Figure 6). The proportions of roe deer increased linearly, with the highest proportions in the years of COVID-19 restrictions: 81.8% in 2020 and 79.1% in 2021. A linear regression explains 96% of the variation. In contrast, the proportions of dogs in the roadkill have been decreasing and were only 3.1 to 4.3% in the years 2020–2022, a decrease in at least three times from the previous year’s value (Figure 6). Linear regression explains 95% of the variation. The proportions of the other three species—moose, wild boar, and red deer—fluctuated close to the mean, at 4.0%, 2.7%, and 0.9%, respectively.

The diversity of roadkill in urbanized areas has followed certain temporal trends. The highest number of roadkilled mammal species was recorded between 2019 and 2022 (Figure 7a), namely, including the two years when COVID-19 restrictions were in place. There were no roadkilled horses in 2020 present for the remaining years 2007–2022, except for 2017. The only year in which three fallow deer individuals were roadkilled in urbanized areas was 2020. Wolves and beavers were also recorded as roadkilled in 2020–2021. Until 2019, there were no roadkilled wolves in urban areas, and one individual was recorded in 2019. Five wolves were roadkilled in 2020 and two in 2021, but in 2022, there were again no road mortalities of this species in urban areas. Of the 16 beavers roadkilled in urban areas, one was recorded in 2007, 2013, 2015, and 2019, two in 2020, four in 2021, and six in 2022, on the assumption that the roadkilling of this species was facilitated by COVID-19 restrictions on mobility. The number of roadkilled roe deer almost doubled compared to 2019, from 700 to 1281–1325 individuals in 2020–2021.

The diversity index was lowest in 2020 and 2021 (Figure 7b), significantly lower than in 2019 (t = 4.54, *p* < 0.0001 and t = 2.29, *p* < 0.025, respectively). The increase in the diversity index in 2022 was significant compared to 2020 (t = 2.97, *p* < 0.005) but not to 2021 (t = 0.61, NS).

The roadkill dominance index was highest in 2020, the first year of the COVID-19 restriction, and was significantly higher than in 2019 (t = 5.87, *p* < 0.0001) and 2021 (t = 2.00, *p* < 0.05). In 2022, it remained the same as in 2021 (Figure 7c).

### 3.3. Monthly Numbers of Roadkilled Mammals on Urban Roads in the Years of COVID-19 Restrictions

In most months when COVID-19 transport and movement restrictions were in place, i.e., April and May 2020, followed by November and December 2020, January to March and June 2021, a higher than expected number of mammal roadkills were observed in urban territories (Figure 8). In numerical terms, the highest excesses over the observed number of roadkills during mobility restrictions occurred in May 2020 (106 roadkills, 90.0%), November 2020 (117 roadkills, 89.6%), December 2020 (104 roadkills, 92.0%), and January 2021 (66 roadkills, 67.8%).

Fewer than expected road mortalities were observed before COVID-19 restrictions in January and February 2020, then in the first month of COVID-19 restrictions, March 2020, with 10 individuals or 16.8%, and near the end of the second restriction period, April and May 2021, with 3 and 8 individuals or 3.7% and 5.4%, respectively. 

The remaining periods in 2020 and 2021, when COVID-19 restrictions were not applied, had a higher than expected number of mammalian roadkills. In 2022, an excess of roadkills was observed in January-April. Later on, the number of mammals killed on urban roads subsequently became lower than expected (Figure 8).

During both COVID-19 restriction periods in Lithuania, the observed number of mammal roadkills in urban areas was 514 individuals higher than expected (χ^2^ = 52.24, *p* < 0.0001), and the medians were different (W = 71, *p* < 0.001). During the first shorter period, the difference was 149 roadkills (χ^2^ = 14.67, *p* < 0.005), while the difference during the second, longer period was 365 roadkills (χ^2^ = 37.57, *p* < 0.0001).

## 4. Discussion

Our study showed that mammalian mortality on urban roads has distinctive characteristics. First, the proportion of domestic animals killed on roads in urbanized areas was found to be 12.2% for the period 2007–2022, significantly higher than on main, national, and regional roads in Lithuania. The proportion of domestic animals in the total roadkill has been steadily decreasing and has been replaced by an increase in the number of roe deer killed on roads. In terms of low numbers of species, low diversity, and the dominance of roe deer in roadkill, urban roads are similar to regional roads.

### 4.1. Roadkill of Domestic Animals

The proportion of domestic animals in roadkill varies from country to country and is higher [26,27] or lower [28] than in Lithuania. In Cyprus, no domestic animal roadkills were recorded [53]. In South Africa, domestic dogs accounted for 11.5% of roadkilled animals, according to Périquet et al. [54]. Texas in the United States was characterized with 31% of domestic animals (like cattle and dogs) in the roadkill of 2010–2016 [55].

The situation is quite different in southern Spain, where domestic animals accounted for more than 95% and dogs alone for more than 80% of all roadkill cases, mainly near urbanized areas [56]. In India, stray dogs are responsible for 69% of accidents [57]. This is in contrast to most of the European countries. In the Czech Republic, for example, according to data from the online system of traffic incidents, domestic animals accounted for 3.17% of all reported roadkills, mostly involving dogs. At the same time, citizen scientists reported that pets accounted for 4.2% of roadkills, but cats were the predominant species [58]. In urbanized areas in Lithuania, dogs accounted for 10% of all roadkilled mammals and were a subdominant species (see Table 1 for numbers). Although the situation in Spain is explained by the high number of stray dogs and the large proportion of farmland habitats [56], in Lithuania, we did not think the same factors were important.

In Japan, the proportion of dogs and cats killed on the roads was equally high regardless of traffic volume [59]. In California, cats were killed on the roads in both roadside and urban landscapes [60]. In New Zealand, the proportion of cats killed on roads declined over time from 1984 to 2005 [61], and the density of cats killed on roads between 2009 and 2014 was 0.90 individuals/100 km [62]. In South Korea, cats dominated among roadkilled domestic animals [27]. In our urban sample, cats accounted for 0.53% of all mammals, but we do not have the ability to estimate the density of roadkilled animals. Nor do we have any evidence of dogs or cats being picked up by their owners, although, after Santos et al. [63], this may be a reason for the carcass persistence after a collision.

Killing an animal on the road can affect driving habits, and vice versa. In Hungary, where 23.9% of drivers surveyed had at least one encounter with domestic animals, emotional restraint was found to be a reason to run over a cat rather than a dog [64]. Spotting both species on the road can be a good reason to slow down [65]. The size of a dog or cat is thought to be enough to cause driver injury [57] or even fatal accidents [66].

### 4.2. Urban Mammals and Roadkills in the Time of COVID-19

In Lithuania, we found that the numbers of roadkills in the urban territories were higher than expected in 2020 and 2021, the years when COVID-19 restrictions on human mobility were in place, but diversity was at its minimum. In the majority of months with COVID-19 restrictions, the observed numbers of mammalian roadkills were significantly higher than those expected according to the long-term trends. This increase is numerically equal to 514 individuals. After the end of the restrictions, mammal roadkill numbers decreased to the expected level. In contrast, the number of mammal roadkills on main, national, and regional roads decreased during the COVID-19 restrictions [30].

The reported results of COVID-19 influence on roadkills are not consistent. Different patterns depend on animal species [67]. No reduction in urban roadkill in Poland during the COVID-19 period has been reported [68]. In Spain, roadkill numbers increased after a short period of reduction [39], in the USA, roadkill numbers decreased after a reduction in traffic volume under COVID-19 restrictions, but the collision rate mostly increased [69].

One of the drivers of roadkill traffic volume was certainly reduced through COVID-19 restrictions [47], but other factors that reflect the situation of animals are not clear. The lockdown did not change some animal behaviors, such as road avoidance, while other attributes, such as spatial distribution, were altered by reduced human mobility [38]. Although species may be adapted to urban environments [20], mammal carnivore records during periods of location should be interpreted with caution [35]. Therefore, we interpret six of the seven known roadkills of grey wolves in urban areas between 2020 and 2021 (see Table 1) as the result of population growth [70]. This may be related to a reduction in hunting pressure, as both restriction periods overlapped with the wolf hunting season, but there is no evidence for this.

There is no doubt that COVID-19 lockdowns made urban environments more attractive for some wildlife species [71], including carnivores such as hyenas [72]. Landscape factors have not changed, just human activity has been significantly reduced [73]. At the same time, during restriction periods, outdoor activities became more extensive than in the pre-COVID-19 period, and appreciation of wildlife, including domestic animals, increased [74]. Recognizing the positive impact of green spaces on urban populations during crises such as the COVID-19 pandemic can contribute to creating urban environments that are more conducive to wildlife in the future [75].

### 4.3. Implications

Urban ecology is important for addressing global biodiversity issues [6]. In urban areas, wildlife is influenced not only by habitat changes, but also by a wide range of human outdoor activities [36]. However, while many drivers of biodiversity change in urban landscapes have recently been analyzed worldwide [3], the influence of COVID-19 is still under investigation. The effects of COVID-19 restrictions on human activities have been similar, but their impacts on wildlife vary. Constraints may therefore contribute to understanding human impacts on biodiversity in the Anthropocene [76]. As a result, many studies have examined the impact of the anthropause, or periods of restriction on human activity around the world.

Understanding the biophilia that results from lockdown can be important for using the human–nature relationship as a tool for nature conservation [77]. In terms of road deaths, more animals indicate more conflict, even if mitigation measures are implemented [78]. Although mitigating roadkill is costly, most measures cannot be implemented in urban environments, with wildlife fencing being the best example. Therefore, WVC can hardly be prevented in urban environments by keeping animals off the roads.

In this context, it is important to determine whether there has been an increase in the number of WVCs in the urbanized areas during the period of the COVID-19 restriction. Our study found that the number of WVCs was 500 cases more than that expected according to the multi-year trend. Such an increase can be expected to have economic and road safety consequences.

As shown by Wilkins et al. [55], in monetary terms, the damage caused by crashes with wild animals is higher than that caused by crashes with domestic animals, with the former still increasing. In Lithuania, the proportion of roe deer killed on urban roads constantly increased between 2007 and 2022, while the proportion of domestic dogs decreased. Car collisions with roe deer, as with other deer, are dangerous due to the size of the animal [79].

Based on the knowledge that wild and domestic mammal roadkill hotspots do not overlap [26], a completely different approach may be needed in urban environments. The best and most cost-effective solution may be to regulate the number of deer [79]. Biodiversity mitigation measures should be included in urban biodiversity management [80], increasing public engagement [11] and the usage of citizen scientists data [81]. This would increase the environmental carrying capacity of urban ecosystems and provide better opportunities for urban wildlife [9].

## 5. Conclusions

The presented results allow for concluding that COVID-19 restrictions on human mobility resulted in an increase in mammalian roadkill on the roads passing through urban areas as well as on their streets. The decrease in roadkill after restrictions was recalled to show how important temporal anthropause was for mammals, their behaviors, and habitat use, allowing for the re-occupation of urban habitats. However, the increase in the roadkill number in urban areas, not corresponding to the reduced transport intensity under COVID-19 restrictions, asks for re-thinking of the mitigation strategies, at least through improving driver awareness on this issue.

## Figures and Tables

**Figure 2 animals-13-03272-f002:**
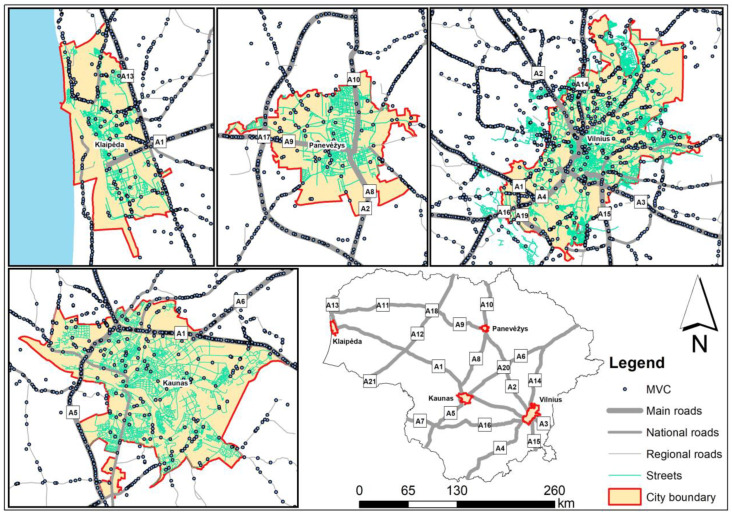
Mammal roadkills (MVC) in the territories of Lithuania’s four largest cities, 2007–2022.

**Figure 3 animals-13-03272-f003:**
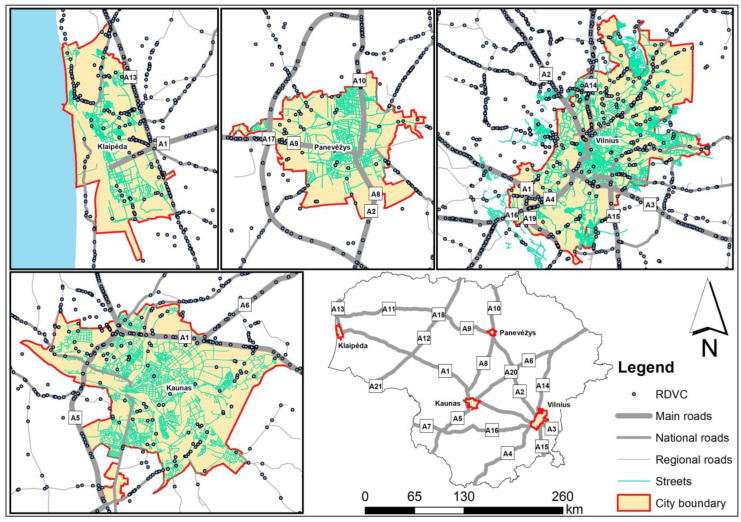
Roe deer roadkills (RDVC) in the territories of Lithuania’s four largest cities, 2007–2022.

**Figure 4 animals-13-03272-f004:**
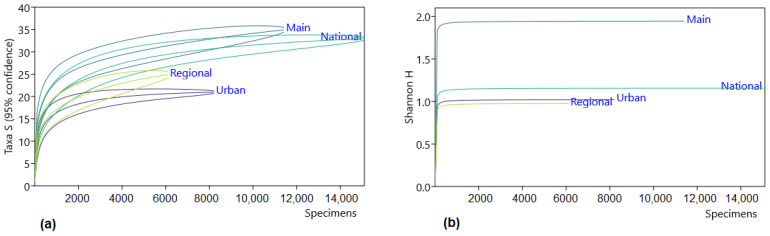
Roadkilled mammal species accumulation curves for Lithuania, 2007–2022, according to road categories: (**a**) number of species, (**b**) diversity based on Shannon’s H. Domestic and wild mammal species data are pooled.

**Figure 5 animals-13-03272-f005:**
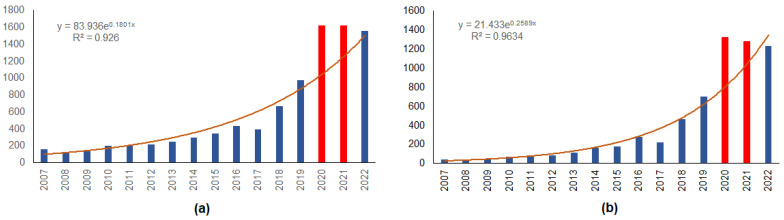
Numbers of roadkilled mammals (**a**) and roe deer (**b**) on the urban roads in Lithuania, 2007–2022. Years with COVID-19 restrictions are denoted by red color. Expected numbers are approximated with exponential regressions.

**Figure 6 animals-13-03272-f006:**
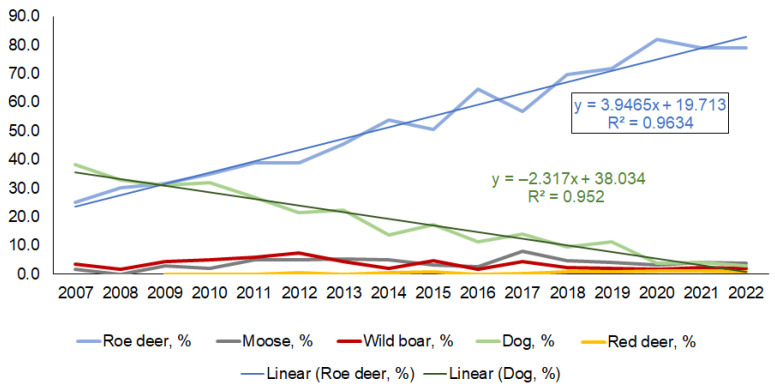
Annual dynamics in the proportions of five dominant roadkilled mammal species in the urbanized territories, 2007–2022. Changes in the proportions of roe deer and dogs are approximated with linear regressions.

**Figure 7 animals-13-03272-f007:**
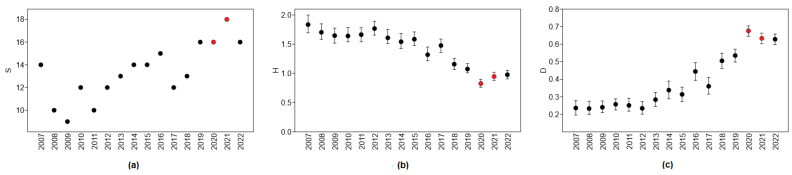
Trends in roadkill diversity: (**a**) number of species, S; (**b**) diversity index, Shannon’s H; (**c**) dominance index, D. Years with COVID-19 restrictions are denoted by red color.

**Figure 8 animals-13-03272-f008:**
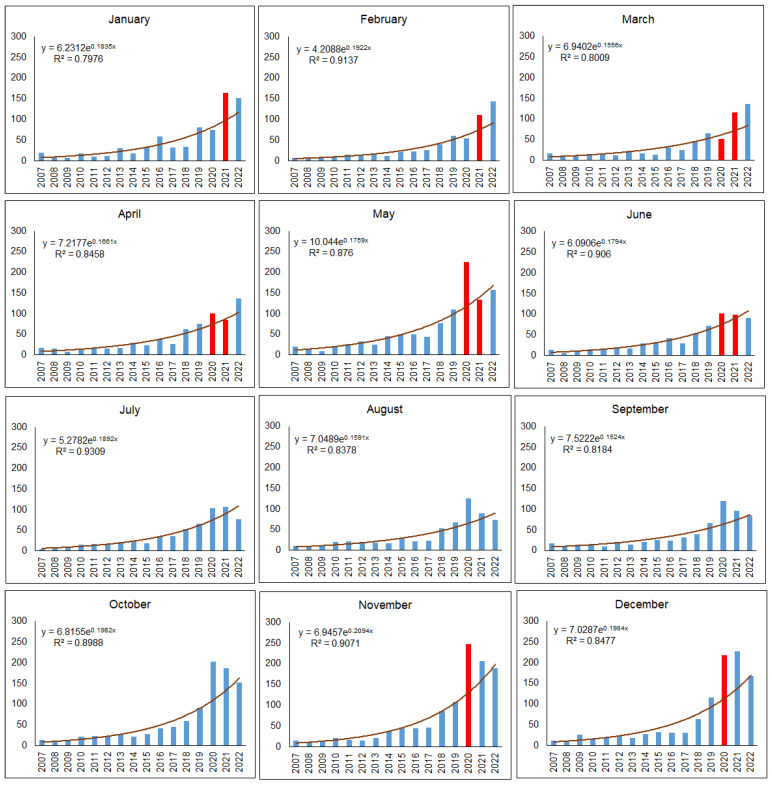
Monthly numbers or mammalian roadkills on urban roads in 2007–2022. Month with COVID-19 restrictions on traffic intensity are denoted by red color.

**Table 1 animals-13-03272-t001:** Species composition and numbers of roadkilled domestic and wild mammals in Lithuania, 2007–2022, by road type.

Species	Total	Urban, *n* (%)	Main, *n* (%)	National, *n* (%)	Regional, *n* (%)
Domestic Mammals					
Horse (*Equus ferus caballus*)	155	59 (38.1)	21 (13.5)	52 (33.5)	23 (14.8)
Cattle (*Bos taurus*)	305	69 (22.6)	53 (17.4)	122 (40.0)	61 (20.0)
Goat (*Capra aegagrus hircus*)	10	4 (40.0)	1 (10.0)	2 (20.0)	3 (30.0)
Sheep (*Ovis aries*)	44	17 (38.6)	7 (15.9)	11 (25.0)	9 (20.5)
Pig (*Sus scrofa domesticus*)	2	1 (50.0)		1 (50.0)	
Rabbit (*Oryctolagus cuniculus*)	5	2 (40.0)	1 (20.0)	1 (20.0)	1 (20.0)
Dog (*Canis familiaris*)	1858	921 (49.6)	345 (18.6)	396 (21.3)	196 (10.5)
Cat (*Felis catus*)	207	49 (23.7)	94 (45.4)	49 (23.7)	15 (7.2)
Wild Mammals					
European bison (*Bison bonasus*)	9			8 (88.9)	1 (11.1)
Moose (*Alces alces*)	2589	366 (14.1)	1017 (39.3)	943 (36.4)	263 (10.2)
Red deer (*Cervus elaphus*)	713	83 (11.6)	244 (34.2)	307 (43.1)	79 (11.1)
Wild boar (*Sus scrofa*)	1962	245 (12.5)	642 (32.7)	746 (38.0)	329 (16.8)
Fallow deer (*Dama dama*)	50	3 (6.0)	10 (20.0)	31 (62.0)	6 (12.0)
Roe deer (*Capreolus capreolus*)	28,024	6303 (22.5)	5316 (19.0)	11,433 (40.8)	4972 (17.7)
Gray wolf (*Canis lupus*)	29	7 (24.1)	15 (51.7)	5 (17.2)	2 (6.9)
Lynx (*Lynx lynx*)	7		3 (42.8)	2 (28.6)	2 (28.6)
Badger (*Meles meles*)	243	21 (8.6)	102 (42.0)	80 (32.9)	40 (16.5)
Red fox (*Vulpes vulpes*)	1214	81 (6.7)	805 (66.3)	260 (21.4)	68 (5.6)
Raccoon dog (*Nyctereutes procyonoides*)	1678	21 (1.3)	1305 (77.8)	294 (17.5)	58 (3.5)
Raccoon (*Procyon lotor*)	10			10 (100.0)	
Eurasian otter (*Lutra lutra*)	22		17 (77.3)	4 (18.2)	1 (4.5)
Marten (*Martes sp.*)	414	1 (0.2)	370 (89.4)	41 (9.9)	2 (0.5)
Pine marten (*Martes martes*)	70		56 (80.0)	14 (20.0)	
Stone marten (*Martes foina*)	35		23 (65.7)	10 (28.6)	2 (5.7)
European polecat (*Mustela putorius*)	188	5 (2.7)	153 (81.4)	30 (16.0)	
American mink (*Neovison vison*)	26		26 (100.0)		
Least weasel (*Mustela nivalis*)	2		2 (100.0)		
Stoat (*Mustela erminea*)	2		1 (50.0)	1 (50.0)	
European hare (*Lepus europaeus*)	691	87 (12.6)	235 (34.0)	242 (35.0)	127 (18.4)
Mountain hare (*Lepus timidus*)	1		1 (100.0)		
Beaver (*Castor fiber*)	65	16 (24.6)	26 (40.0)	17 (26.2)	6 (9.2)
Muskrat (*Ondatra zibethicus*)	2		2 (100.0)		
Red squirrel (*Sciurus vulgaris*)	40		34 (85.0)	6 (15.0)	
Norway rat (*Rattus norvegicus*)	10		5 (50.0)	4 (40.0)	1 (10.0)
Black rat (*Rattus rattus*)	1		1 (100.0)		
Yellow-necked mouse (*Apodemus flavicollis*)	2		2 (100.0)		
Bank vole (*Clethrionomys glareolus*)	3			3 (100.0)	
Eastern hedgehog (*Erinaceus concolor*)	1158	8 (0.7)	953 (82.3)	182 (15.7)	15 (1.3)
European mole (*Talpa europaea*)	20		11 (55.0)	7 (35.0)	2 (10.0)
Water shrew (*Neomys fodiens*)	1		1 (100.0)		
Common shrew (*Sorex araneus*)	2			2 (100.0)	
Unknown	5685	823 (14.5)	1836 (32.3)	2200 (38.7)	826 (14.5)

**Table 2 animals-13-03272-t002:** Diversity indices of roadkilled mammals in Lithuania, 2007–2022, by road type. Superscript letters denote differences, significant at *p* < 0.05.

Index	Total	Urban	Main	National	Regional
Number of Individuals, n	47,554	9192	13,736	17,516	7110
Number of Species, S	40	21 ^a^	35 ^b^	33 ^b^	25 ^a^
Dominance, D	0.47	0.58 ^b^	0.25 ^c^	0.57 ^b^	0.63 ^a^
Diversity, Shannon’s H	1.41	1.02 ^a^	1.95 ^b^	1.16 ^c^	0.98 ^a^

## Data Availability

Due to the ongoing investigation, data of this study are available from the corresponding author upon personal request.

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
