# Peer review of "Mammal Roadkills in Lithuanian Urban Areas: A 15-Year Study"

_animals, 2023, doi:10.3390/ani13203272_

Round 1
Reviewer 1 Report
The authors present the analyses of roadkills on different highways at the national level in Lithuania. The data set is quite substantial, and the analyses are appropriate. The presentation is less so, and the text needs to be edited.
The paper is of great interest. I encourage the authors to undertake a major revision of their presentation and to improve the flow of the text by having it edited for language.
Specifically:
I think the authors should highlight more the effects of the COVID-19 lockdowns with a stress on the species that turn up in the samples – notably wolves and beavers. In general, I think the paper will benefit from comparing before, during, and after to a much greater degree than is at present. Include this also in the title.
Also, I think the fact that the authors chose not to present a map because some of the samples are vague is a mistake. That is badly needed to visualize where the “red” roads are, even if they have to average the location of some points or remove them from the analyses.
Table 1 – add columns of %
The authors might also consider translating the numbers into biomass. Think this will also contribute to understanding fatalities.
Please do not start a sentence with a number – Pg 9, line 272: 2020 was ……
Fig 6 is overwhelming and makes it difficult to compare. Recommend authors to make them seasonal, limiting to four graphs.
The references are missing some recent papers about roadkills:
Swinnen et al. 2022. Nature Conservation 47: 121–153.
doi: 10.3897/natureconservation.47.72970
Hadad et al. 2023. Ecological Research 38:664-675. doi: 10.1111/1440-1703.12399.
A subject relevant to implications that have been ignored and should be included:
Bujoczek et al. 2011. Biological Conservation 144:1036-1039. DOI: 10.1016/j.biocon.2010.12.022
Improve the flow of the text by having it edited for language and shortening some of the very wordy sentences and paragraphs.
Author Response
Rev#1 comments and answers
Comment: The authors present the analyses of roadkills on different highways at the national level in Lithuania. The data set is quite substantial, and the analyses are appropriate. The presentation is less so, and the text needs to be edited. The paper is of great interest. I encourage the authors to undertake a major revision of their presentation and to improve the flow of the text by having it edited for language.
Answer: thank you. Where possible, we did revision according your comments. On the rest, please find explanations below.
Specifically:
Comment: I think the authors should highlight more the effects of the COVID-19 lockdowns with a stress on the species that turn up in the samples – notably wolves and beavers. In general, I think the paper will benefit from comparing before, during, and after to a much greater degree than is at present. Include this also in the title.
Answer: There isn't a lot to discuss on this matter since the incidence of roadkill for these two species in urban environments remains relatively low. Additional text has been included:
” Until 2019, there were no roadkilled wolves in urban areas, and one individual was recorded in 2019. Five wolves were roadkilled in 2020 and two in 2021, but in 2022 there were again no road mortalities of this species in urban areas. Of the 16 beavers roadkilled in urban areas, one was recorded in 2007, 2013, 2015 and 2019, two in 2020, four in 2021 and six in 2022, on the assumption that the roadkilling of this species on roads was facilitated by COVID-19 restrictions of mobility.
Comment: Also, I think the fact that the authors chose not to present a map because some of the samples are vague is a mistake. That is badly needed to visualize where the “red” roads are, even if they have to average the location of some points or remove them from the analyses.
Answer: We add two new figures per your comment, presenting geocoded mammal roadkills and roe deer roadkills in the four biggest urban territories of Lithuania. All roadkills in the country are given also, as Appendices, one for all mammals, second for roe deer.
Comment: Table 1 – add columns of %
Answer: Proportions for the species roadkills have been included in Appendix A. However, introducing four additional columns with percentages would necessitate shifting Table 1 into a landscape format. Furthermore, text concerning these proportions has been incorporated:
Only a few species are more likely to be killed on urban roads than on other types of roads - main, national or regional. All of these species - horse, goat, rabbit and dog - were domestic animals (Table A1)
Comment: The authors might also consider translating the numbers into biomass. Think this will also contribute to understanding fatalities.
Answer: While your suggestion is indeed commendable, we need to clarify that our current analysis does not encompass human fatalities, as these are designated for examination in a distinct research paper. We do intend to incorporate the body mass of road-killed animals in cases involving human fatalities and injuries. Within the framework of our present manuscript's objectives, however, assessing biomass falls beyond our scope. On a related note, the reports contain a multitude of 'not recorded' factors, such as missing data on animal gender and age, which lead to exceedingly ambiguous generalizations.
Comment: Please do not start a sentence with a number – Pg 9, line 272: 2020 was ……
Answer: changed as advised
Comment: Fig 6 is overwhelming and makes it difficult to compare. Recommend authors to make them seasonal, limiting to four graphs.
Answer: When creating seasonal graphs, we would be blending data from months with COVID restrictions and those without restrictions in varying proportions. Consequently, seasonal charts could potentially obscure the patterns that are presently evident.
It's important to note that even during periods of restrictions, the number of animal accidents varied from month to month. Combining the data into 'COVID restriction' versus 'no restriction' periods might result in the loss of valuable information.
Comment: The references are missing some recent papers about roadkills:
Swinnen et al. 2022. Nature Conservation 47: 121–153. doi: 10.3897/natureconservation.47.72970
Hadad et al. 2023. Ecological Research 38:664-675. doi: 10.1111/1440-1703.12399.
A subject relevant to implications that have been ignored and should be included: Bujoczek et al. 2011. Biological Conservation 144:1036-1039. DOI: 10.1016/j.biocon.2010.12.022
Answer: thank you, Swinnen and Hadad now cited in Discussion, however Bujoczek’s paper is about the bird roadkills, which we do not analyze.
Comment: Improve the flow of the text by having it edited for language and shortening some of the very wordy sentences and paragraphs.
Answer: we did our best, largely rewriting Simple summary and Abstract. Rest of the text was approved by Rev#2, but changes were still done, mainly for clarity and shortening of long sentences.
Reviewer 2 Report
Thank you for the opportunity to review the manuscript. The presented paper covered the topic of wildlife-vehicle collisions in Lithuania for 15 years. The Authors showed the pattern and changes in collisions for wild and domestic species of animals and put light on the topic of COVID-19 pandemic's influence on wildlife-vehicle collisions.
The first part of the manuscript (abstract and simple summary) must be rewritten, but I find the other part of the paper easy to follow and understandable. Detailed comments are below.
Simple summary – please rewrite this chapter. It’s hard to follow in that version. Moreover, the language could be improved here.
l. 10 What do you mean by “which types of animals”? Species?
l. 11 Please add the Latin name of roe deer
l.12 “hit in urban areas, as they are on normal roads” – there are no normal roads in urban areas of Lithuania? I don’t understand that.
l. 13 I think two sentences are connected here – please clarify it
l.14 “During the 2020 and 2021 COVID roadblocks in urban areas, (…)” – please add a verb
l. 15 What do you mean by “it”?
Abstract
l. 19 Remove “such”
l.21/22 Trends in what, changes of what? Please rewrite the aim of the study in the abstract. The aim of the study should be easy to understand, although based on that sentence, I don’t understand the paper's goal.
In my opinion in abstract, there are too many results of COVID-19 influence on wildlife-vehicle collisions. There is a lack of results on trends, and changes from 15 years of collected data.
Keywords – I suggest “urban areas”
l. 37 factor of what
Reading the other part of the manuscript I feel like other people wrote it than abstract and simple summary.
l. 69 Please add references after “measures”
l. 117 “out of these, 1,750.71 km” – please remove the comma
l. 121 “data is not available” – please change “is” into “are”
l. 200-201 “Values, 200 denoted by different letters, differ at p < 0.05 or higher” - Please write specifically
l. 226 What do you mean by “National roads saw”?
I wish the Authors all the best.
Language must be improved in the Abstract and Simple Summary.
Author Response
Rev#2 comments and answers
Comment: Thank you for the opportunity to review the manuscript. The presented paper covered the topic of wildlife-vehicle collisions in Lithuania for 15 years. The Authors showed the pattern and changes in collisions for wild and domestic species of animals and put light on the topic of COVID-19 pandemic's influence on wildlife-vehicle collisions.
Answer: we appreciate your review, it is straightforward and helpful.
Comment: The first part of the manuscript (abstract and simple summary) must be rewritten, but I find the other part of the paper easy to follow and understandable. Detailed comments are below.
Simple summary – please rewrite this chapter. It’s hard to follow in that version. Moreover, the language could be improved here.
- 10 What do you mean by “which types of animals”? Species?
Answer: we meant “which species of wild and domestic mammals”
- 11 Please add the Latin name of roe deer
Answer: added
l.12 “hit in urban areas, as they are on normal roads” – there are no normal roads in urban areas of Lithuania? I don’t understand that.
Answer: apologies for language. We meant, that roe deer was most numerous species killed on urban and non-urban roads.
- 13 I think two sentences are connected here – please clarify it
Answer: we rewrote this sentence
l.14 “During the 2020 and 2021 COVID roadblocks in urban areas, (…)” – please add a verb
- 15 What do you mean by “it”?
Answer: we rewrote this part
Abstract
- 19 Remove “such”
Answer: removed
l.21/22 Trends in what, changes of what? Please rewrite the aim of the study in the abstract. The aim of the study should be easy to understand, although based on that sentence, I don’t understand the paper's goal.
In my opinion in abstract, there are too many results of COVID-19 influence on wildlife-vehicle collisions. There is a lack of results on trends, and changes from 15 years of collected data.
Answer: we did our best in rewriting both Simple summary and abstract according your comments.
Comment: Keywords – I suggest “urban areas”
Answer: included
Reading the other part of the manuscript I feel like other people wrote it than abstract and simple summary.
Answer: these “other people” might have been only co-authors of this paper, no one else J.
Comment: l. 37 factor of what
Answer: misunderstanding cleared
Comment: l. 69 Please add references after “measures”
Answer: it is the same [22]. To clarify, we joined two sentences into one.
Comment: l. 117 “out of these, 1,750.71 km” – please remove the comma
Answer: removed
Comment: l. 121 “data is not available” – please change “is” into “are”
Answer: changed as advised
Comment: l. 200-201 “Values, 200 denoted by different letters, differ at p < 0.05 or higher” - Please write specifically
Answer: to avoid unnecessary mess with all differences, we excluded words “or higher”; now it should be clear, that all differences are significant at p < 0.05.
Comment: l. 226 What do you mean by “National roads saw”?
Answer: apologies, misunderstanding cleared. Now sounds “On national roads, the moose (5.4%) and the wild boar (4.3%) were prevalent species, while regional roads mirrored the same pattern, with these species comprising 3.7% and 4.6%, respectively.”
Comments on the Quality of English Language: Language must be improved in the Abstract and Simple Summary.
Answer: we had native speaker check on the language after revision. Thank you, we fully agree that Simple summary and Abstract required rewriting.
Round 2
Reviewer 1 Report
Animals-2648290 Mammal Roadkills in Lithuanian Urban Areas: A 15-Year Study. Authors: Linas Balčiauskas *, Andrius Kučas , Laima Balčiauskienė
This is a revision of an earlier version submitted to Animals. The authors present the analyses of roadkills in Lithuania and distinguish between domestic and wild species, the anthropopause, and the outside. The data is important, and the article is certainly worthy of future publication after the authors have fully addressed the reviewers’ comments.
The answer to adding % columns to the table just because it is inconvenient because the format needs to be landscape is not acceptable. The % can be added to the present table in parenthesis () beside each number. I still think this is an important point and the spreadsheet can be submitted as a separate file if necessary. The data presented in the text is lost in the fuzziness of the words and is not always immediately apparent to the reader.
Furthermore, the authors ignored Bujoczek et al. 2011. (DOI: 10.1016/j.biocon.2010.12.022) just because it refers to birds does not mean it’s scientifically sound. A published work is cited not only because it belongs to a particular taxon but also because of the ecological or evolutionary principles mentioned in it. I still believe that the authors need to address the issue of the evolutionary consequences of accidental roadkills, particularly on wildlife populations.
I think the authors have taken a minimalist approach to the revision. The flow of language is still problematic; although the authors note that reviewer 2 had no problem with it. I am including an example of my editing of the abstract so that authors understand that they need to have the text properly edited before resubmitting.
We investigated roadkillsexamined traffic fatalities in urban areas in of Lithuania from 2007 to 2022, including two periods with of COVID-19 restrictions on human people’s movement. We analyzed the proportions of wild and domestic animals in roadkills, annual trends, the predominant species involved and monthly changes during the restrictions. Urban roads were characterized by a low species biodiversity of road-killed mammals, with a dominance of roe deer (Capreolus capreolus) dominating. The total numbers increased exponentially during the study period. At 12.2%, The the proportion of domestic animals, 12.2%, was significantly exceeded higher that than on non-urban roads in the country. The proportion of domestic animals decreased from over 40% in 2007–2009 to 3.7–5.4% in 2020–2022, while the proportion of wild mammals increased from 36.1–39.6% to 89.9–90.6% respectively. During the periods of COVID-19 restrictions, the number of roadkills in urban areas was significantly higher than expected on the basis based of on long-term trends. Compared to 2019, the number of road-killed roe deer in 2020–2021 almost doubled from 700 to 1,281–1,325 individuals. These However, these anthropause effects were , however, temporary in nature. The disbalance imbalance between the number of roadkills number and transport traffic intensity might may require new mitigation strategies to sustain conserve mammal populations in urban areas, at least through by improving driver driver-awareness on of the issueproblem.
Throughout text add comma after i.e.
Line 181 – do not start a sentence with “E.g., ….” Write out For example …
Line 324 – change header to “Data Analysis”
Line 359 – you have included rabbits in domestic animals. Is this correct?
Lines 404, 405 – Legend Table 2. I did not understand the reason behind the different letters. If all “differ at p < 0.05.” then why give different letters? Either specify in the legend or all with an asterisk.
Table 1, Line 467 – raccoon dog (Nyctereutes procyonoides)? I thought these exist only in Japan and the Far East. Please recheck the correct name for the species involved. Are they an invasive species to Eastern Europe? Lithuania?
Fig 8 (previously Fig 6) – if you really want to present monthly graphs, you should keep the values on the Y-axis the same throughout so that the reader gets a correct impression of the differences between months.
Line 728 - In the Czechia,…. – rephrase correctly
Lines 898-899 - Your final sentence is not based on your data as you have not assessed transport intensity. Rephrase the sentence.

See attached file:
I am including an example of my editing of the abstract so that authors understand that they need to have the text properly edited before resubmitting.
We investigated roadkillsexamined traffic fatalities in urban areas in of Lithuania from 2007 to 2022, including two periods with of COVID-19 restrictions on human people’s movement. We analyzed the proportions of wild and domestic animals in roadkills, annual trends, the predominant species involved and monthly changes during the restrictions. Urban roads were characterized by a low species biodiversity of road-killed mammals, with a dominance of roe deer (Capreolus capreolus) dominating. The total numbers increased exponentially during the study period. At 12.2%, The the proportion of domestic animals, 12.2%, was significantly exceeded higher that than on non-urban roads in the country. The proportion of domestic animals decreased from over 40% in 2007–2009 to 3.7–5.4% in 2020–2022, while the proportion of wild mammals increased from 36.1–39.6% to 89.9–90.6% respectively. During the periods of COVID-19 restrictions, the number of roadkills in urban areas was significantly higher than expected on the basis based of on long-term trends. Compared to 2019, the number of road-killed roe deer in 2020–2021 almost doubled from 700 to 1,281–1,325 individuals. These However, these anthropause effects were , however, temporary in nature. The disbalance imbalance between the number of roadkills number and transport traffic intensity might may require new mitigation strategies to sustain conserve mammal populations in urban areas, at least through by improving driver driver-awareness on of the issueproblem.
Author Response
Rev#1 round 2 Comments and Suggestions for Authors
Animals-2648290 Mammal Roadkills in Lithuanian Urban Areas: A 15-Year Study. Authors: Linas Balčiauskas *, Andrius Kučas , Laima Balčiauskienė
This is a revision of an earlier version submitted to Animals. The authors present the analyses of roadkills in Lithuania and distinguish between domestic and wild species, the anthropopause, and the outside. The data is important, and the article is certainly worthy of future publication after the authors have fully addressed the reviewers’ comments.
Comment: The answer to adding % columns to the table just because it is inconvenient because the format needs to be landscape is not acceptable. The % can be added to the present table in parenthesis () beside each number. I still think this is an important point and the spreadsheet can be submitted as a separate file if necessary. The data presented in the text is lost in the fuzziness of the words and is not always immediately apparent to the reader.
Answer: We did as required, Table A1 removed from Appendix.
Comment: Furthermore, the authors ignored Bujoczek et al. 2011. (DOI: 10.1016/j.biocon.2010.12.022) just because it refers to birds does not mean it’s scientifically sound. A published work is cited not only because it belongs to a particular taxon but also because of the ecological or evolutionary principles mentioned in it. I still believe that the authors need to address the issue of the evolutionary consequences of accidental roadkills, particularly on wildlife populations.
Rebuttal: the omission of the article in question has nothing to do with its scientific validity. We will not cite this article because (a) we are not writing about birds, (b) we are not writing about the body quality of dead mammals either, and (c) we are not dealing with the evolutionary aspects of roadkills. The article we are forced to cite is completely out of our scope.
Comment: I think the authors have taken a minimalist approach to the revision. The flow of language is still problematic; although the authors note that reviewer 2 had no problem with it. I am including an example of my editing of the abstract so that authors understand that they need to have the text properly edited before resubmitting.
Answer: Concerning the suggested changes to the English, we disagree with most of the proposed changes. For example, we did not examine road fatalities (=persons killed on roads), but as we said roadkills (=animals killed in road accidents). Elsewhere, we have a native speaker who assures us that most of the suggested changes are simply rephrasing one perfectly acceptable phrase to another acceptable phrase. They also wanted us to change 'periods of restrictions' to period of restriction' ...wrong, there were two periods. We will make minor changes where we feel appropriate.
Comment: Throughout text add comma after i.e.
Answer: added in Line 320, no more such cases
Comment: Line 181 – do not start a sentence with “E.g., ….” Write out For example …
Answer: it was Line 81, changed as required
Comment: Line 324 – change header to “Data Analysis”
Answer: if you mean Line 166, we changed header as requested; there are no header in Line 324
Comment: Line 359 – you have included rabbits in domestic animals. Is this correct?
Answer: absolutely correct. In Lithuania there are no wild rabbits
Comment: Lines 404, 405 – Legend Table 2. I did not understand the reason behind the different letters. If all “differ at p < 0.05.” then why give different letters? Either specify in the legend or all with an asterisk.
Rebuttal: No, not all differences are significant. Let me explain on example:
Number of species, S |
40 |
21 a |
35 b |
33 b |
25 a |
21 is not differen from 25; but both differ significantly from 33 and 35. 33 did not differ from 35. Asterisks will not help here.
Comment: Table 1, Line 467 – raccoon dog (Nyctereutes procyonoides)? I thought these exist only in Japan and the Far East. Please recheck the correct name for the species involved. Are they an invasive species to Eastern Europe? Lithuania?
Rebuttal: you are wrong here, raccoon dog (Nyctereutes procyonoides) now is inhabiting all Europe, and is invasive species everywhere. Name of species is correct.
Comment: Fig 8 (previously Fig 6) – if you really want to present monthly graphs, you should keep the values on the Y-axis the same throughout so that the reader gets a correct impression of the differences between months.
Answer: We did requested change, however, idea was to see differences between years of the same month, thus, scale was not so significant.
Line 728 - In the Czechia,…. – rephrase correctly
Answer: We rephrased sentence with changing country name.
However, if you mean country name, then in 2022, the American AP Stylebook stated in its entry on the country that "Czechia, the Czech Republic. Both are acceptable. The shorter name Czechia is preferred by the Czech government. If using Czechia, clarify in the story that the country is more widely known in English as the Czech Republic. The same name used in EU country profiles. In 2016, the Czech government requested that "Czechia" be recognized as the official one-word name for the country in international use to provide a shorter and more straightforward alternative to "Czech Republic.
Comment: Lines 898-899 - Your final sentence is not based on your data as you have not assessed transport intensity. Rephrase the sentence.
Answer: Lines 898–899 are non existent. So we presume, these are Lines 456–458. Yes, we have no detail data on transport intensity during COVID restrictions, Even so, we have Bates et al. reported a 36% reduction in average driving time in Lithuania [47].
We are therefore absolutely convinced that we have the right to talk about the mismatch between the intensity of animal deaths in the city and the traffic flows. As there are currently no other mitigation measures available in urban areas, informing motorists would be the only option.
Comments on the Quality of English Language, See attached file: peer-review-32839700.v1.pdf
Comment: I am including an example of my editing of the abstract so that authors understand that they need to have the text properly edited before resubmitting.
Answer: Concerning the suggested changes to the English, we disagree with most of the proposed changes. For example, we did not examine road fatalities (=persons killed on roads), but as we said roadkills (=animals killed in road accidents). Elsewhere, we have a native speaker who assures us that most of the suggested changes are simply rephrasing one perfectly acceptable phrase to another acceptable phrase. They also wanted us to change 'periods of restrictions' to period of restriction' ...wrong, there were two periods. We will make minor changes where we feel appropriate.